# Translational Aspects in Metaplastic Breast Carcinoma

**DOI:** 10.3390/cancers16071433

**Published:** 2024-04-07

**Authors:** Elizve Nairoby Barrientos-Toro, Qingqing Ding, Maria Gabriela Raso

**Affiliations:** 1Department of Translational Molecular Pathology, The University of Texas MD Anderson Cancer Center, Houston, TX 77030, USA; enbarrientos@mdanderson.org; 2Department of Anatomical Pathology, The University of Texas MD Anderson Cancer Center, Houston, TX 77030, USA; qqding@mdanderson.org

**Keywords:** metaplastic breast carcinoma, triple-negative breast carcinoma, tumor immune microenvironment, epithelial–mesenchymal transition, clinical trials

## Abstract

**Simple Summary:**

Breast cancer is the most common cancer among women. Metaplastic breast carcinoma is a rare, heterogeneous group of invasive breast carcinomas that are more aggressive, poorly responsive to neoadjuvant chemotherapy, and have higher rates of chemoresistance than most other breast tumors. This article aims to provide an overview and summary of the current knowledge regarding the translational aspects of metaplastic breast carcinoma that could open the possibility of discovering specific treatment targets and knowing prognostic factors.

**Abstract:**

Breast cancer is the most common cancer among women. Metaplastic breast carcinoma (MpBC) is a rare, heterogeneous group of invasive breast carcinomas, which are classified as predominantly triple-negative breast carcinomas (TNBCs; HR-negative/HER2-negative). Histologically, MpBC is classified into six subtypes. Two of these are considered low-grade and the others are high-grade. MpBCs seem to be more aggressive, less responsive to neoadjuvant chemotherapy, and have higher rates of chemoresistance than other TNBCs. MpBCs have a lower survival rate than expected for TNBCs. MpBC treatment represents a challenge, leading to a thorough exploration of the tumor immune microenvironment, which has recently opened the possibility of new therapeutic strategies. The epithelial–mesenchymal transition in MpBC is characterized by the loss of intercellular adhesion, downregulation of epithelial markers, underexpression of genes with biological epithelial functions, upregulation of mesenchymal markers, overexpression of genes with biological mesenchymal functions, acquisition of fibroblast-like (spindle) morphology, cytoskeleton reorganization, increased motility, invasiveness, and metastatic capabilities. This article reviews and summarizes the current knowledge and translational aspects of MpBC.

## 1. Introduction

According to the American Cancer Society, the estimated breast cancer cases and deaths in women, in the United States, by the year 2024, are 32% and 21%, respectively [1]. Invasive breast cancer is a heterogeneous group of malignancies with estimated new cases and deaths among US women in 2022 of 287,850 and 43,250, respectively [2,3]. Based on the published and registered data, Arnold et al. have estimated a 40% increase in the number of newly diagnosed cases by 2040 and a 50% increase in deaths from breast cancer worldwide [4]. Metaplastic breast carcinomas (MpBCs) comprise a heterogeneous group of rare invasive breast carcinomas that are classified by their molecular profile as predominantly triple-negative breast carcinomas (TNBCs). TNBCs account for 10–20% of all breast carcinomas. As a group, they show heterogeneity, with different clinical–histologic features, genetic–molecular alterations, and responses to treatment compared with other types of breast carcinoma. According to the American Cancer Society, triple-negative cases were present in 10% of all racial groups among those diagnosed with invasive female breast cancer in the United States between the years 2015 and 2019 [5]. Although worldwide statistics reflect that MpBCs (as defined by the World Health Organization [WHO]) represent less than 1% of all invasive breast carcinomas [6], some authors consider that up to 5% of any breast carcinoma may experience some metaplastic change toward a different growth pattern, especially toward a non-glandular pattern [7].

Immunohistochemistry is an essential tool for achieving a correct diagnosis, defining each histological type, as well as differentiating it from other tumors with similar histological characteristics [8,9,10,11]. MpBCs’ molecular profile classified them as predominantly TNBCs [11,12]. The WHO Classification of Tumors of the Breast defines them as a group of neoplasms “characterized by differentiation of the neoplastic epithelium into squamous cells and/or mesenchymal-looking elements”. The sarcomatous components include spindle, chondroid, osseous, and rhabdomyoid differentiation. These entities could be composed entirely by metaplastic elements, or by a mixture of carcinoma and metaplastic areas. Both components (carcinomatous and sarcomatous) could have a clonal origin [13,14,15,16,17,18].

This article aims to provide an overview and summary of the current knowledge regarding the translational aspects of MpBC.

## 2. Clinical–Pathologic Features

Patients with MpBC are typically women between 50 and 60 years old [19,20]. The tumors show similar clinical features to ER-negative invasive breast carcinoma of no special type (NST) [6]. Mammography complemented with ultrasonography usually reveals a mass corresponding to a palpable lump. Calcifications are uncommon; when present, a ductal carcinoma in situ and/or osseous differentiation should be investigated as a part of the lesion. These tumors are hard to the touch, show a multinodular, grey-white, and well-circumscribed macroscopic appearance [21].

In the series reviewed by González-Martínez et al., MpBCs are likely to present at an advanced stage (stage II) [22]. Clinically, the size of these tumors varies from 2 to >10 cm, metastases are more frequent, and these tumors have chemoresistance potential [10,22,23]. Moreover, they found about 35% of the cases demonstrated positive lymph nodes, 13% showed spreading to visceral organs at the time of diagnosis, and approximately 20% showed lymphovascular invasion.

## 3. Histopathological and Immunohistochemical Features

In the context of MpBCs’ histopathological and immunohistochemical features, the 2019–2020 WHO Classification of Tumors Editorial Board emphasized the importance of a descriptive classification system [6]. This classification identifies six distinct histological subtypes:Low-grade adenosquamous carcinoma (LGASC);Fibromatosis-like metaplastic carcinoma (FLMC);Squamous cell carcinoma (SqCC) (Figure 1 and Figure 2);Spindle cell carcinoma (SpCC) (Figure 3);Metaplastic carcinoma with heterologous mesenchymal differentiation (MCHMD);Mixed metaplastic carcinoma (MMC), where the tumor exhibits mixed components.

This system is primarily based on the specific metaplastic elements present within the tumors, while also highlighting the potential overlap that can exist between various histological subtypes [6]. Consequently, numerous cases diagnosed as MpBCs, are accompanied by a detailed description of the histological component or a combination of different component types and/or differentiations. The following figures illustrates a few examples: spindle cell carcinoma component with pleomorphic features (Figure 4), metaplastic breast carcinoma with chondroid differentiation (Figure 5), metaplastic breast carcinoma with a matrix-producing component (Figure 6), metaplastic breast carcinoma with osseous differentiation (Figure 7), and extensive trabeculae bone and hematopoietic tissue (Figure 8).

It is worth mentioning that LGASC and FLMC are considered low-grade subtypes, while all other subtypes fall under the category of high-grade MpBCs, which is associated with a poorer prognosis [6,24,25,26]. The heterogeneity that characterizes MpBCs is evident not only in the histological variations, but also in the expression patterns revealed through immunohistochemistry. Moreover, there is a wide array of potential differential diagnoses to be considered within each subtype, which extends to the outcomes and prognosis. This complexity underscores the necessity for a precise diagnosis in each case [27,28]. Immunohistochemistry is considered an essential diagnostic method for MpBCs. Notably, more than 90% of MpBCs lack expression of estrogen receptor (ER), progesterone receptor (PR), and ERBB2 (HER2), classifying them as TNBC [9,29,30,31].

MpBCs, in line with their molecular classification, demonstrate high-molecular-weight cytokeratins/basal markers, including CK5/6 and CK34BE12. Additionally, they may exhibit staining with broad-spectrum cytokeratins [32]. The marker p63, known for its high sensitivity and specificity [31], plays a crucial role in diagnosing these tumors, showing staining in both the epithelial and spindle cell components [33]. CK7 demonstrates positivity in roughly 30–60% of MpBCs. Furthermore, myoepithelial markers such as smooth muscle actin (SMA), CD10, and maspin are frequently expressed, with CD10 being more commonly observed in SpCC than in other variants [34]. E-cadherin (CDH1) may be expressed abnormally, and EGFR is often overexpressed [35].

Recent studies have highlighted the utility of TRPS1 in effectively distinguishing between MpBC and primary sarcomas [36]. In a recent comparative analysis with GATA3 and SOX10, it was observed that TRPS1 exhibited a higher expression in MpBC cases (95% for TRPS1 versus 50% and 49% for GATA3 and SOX10, respectively) [37]. These findings underscore the potential of TRPS1 as a valuable marker in MpBC differential diagnosis.

A few negative markers can help diagnose MpBCs. MpBCs do not express CD34, which helps distinguish them from other spindle cell lesions. They also tend to lack expression of desmin and SMMHC.

Numerous authors recommend a combination of various stains based on histological features, especially in cases displaying spindle cell or sarcomatoid morphology [31,38]. Additionally, the choice of sample type, core, or surgical biopsy should be carefully considered [29,32].

## 4. Molecular Aspects in Metaplastic Breast Carcinoma: Associated Mutations, Involved Pathways, and Transcriptomic Features

MpBCs’ molecular characterization, as well as their immunohistochemical profile, is consistent with the TNBC group. Numerous studies have indicated that a high percentage of TNBCs show basal-like features or claudin-low molecular subtypes [39,40,41]. In addition, a low percentage of them have different morphology, biological behavior [16], and genetic alterations.

Further subclassification of these tumors grouped them under the mesenchymal-like molecular subtype of TNBC. Despite belonging to this group, MpBCs seem to be more aggressive and poorly responsive to neoadjuvant chemotherapy. Additionally, MpBCs have been found to have a lower survival rate than other TNBCs.

Lien et al., in 2007, used the significance analysis of microarrays (SAM) to compare the molecular characteristics of MpBCs with those of ductal breast carcinomas [42]. They found a total of 208 genes, most of which were underexpressed and downregulated in MCBs and were related to maintaining an epithelial phenotype. On the one hand, several overexpressed genes had an association with extracellular matrix (ECM), including genes related to skeletal development and/or chondro-ossification. On the other hand, *ESR1* and *ERBB2* genes were downregulated, a clear reflection of the lack of expression of ER and HER-2/neu in MpBCs.

Most series examining MpBC patient samples have determined that *TP53* is the most frequently mutated gene, and, in some series, all histological subtypes were affected by this mutation [22], whereas other studies have demonstrated variations. For example, SqCC and MCHMDs had more frequent *TP53* mutations, but none of the pure SpCC had *TP53* mutations [43]. The second most frequently mutated gene is *PIK3CA* [44].

The *TERT* promoter gene, which is related to telomerase activation and also activates the WNT pathway, was mutated in 25% of MpBCs evaluated in one series [43]. The authors of that report also found that this mutation was present in tumors with squamous and spindle cell differentiation but was not found in matrix-producing carcinomas. Tray et al. found that *TERT* genomic alterations ranked fourth in their results [45].

Other investigators have found different mutated gene frequencies. For example, Ng et al. described *TTN* mutated in 31%, and Hayes et al. found *CTNNB1* mutated in 25.9% of total MpBCs [46,47].

Other mutations found in some MpBCs are *PTEN*, *PIK3R1*, *NF1*, *HRAS*, and *AKT1*. Authors have also demonstrated that different types of mutation can occur in different genes that belong to the same signaling pathway [48,49].

The *NF1* gene, belonging to the PI3K/AKT signaling pathway, shows somatic mutations in MpBC, especially when a spindle cell component and squamous component is present [22].

*MYC* is the most amplified gene, followed by *EGFR*. Their amplification has been found more frequently in tumors with squamous or spindle differentiation [35]. *CDKN2A*/*CDKN2B* is the most common gene loss, followed by *PTEN* [22]. Genes involved in DNA repair have also been studied in MpBC and found to be downregulated compared to other TNBCs. These genes include *BRCA1*, *BRCA2*, and *ATM* [41,50,51,52,53].

Appendix A presents the most frequently mutated genes in MpBC according to the number of series that described them. Appendix A present mutated genes in MpBC found in a single series.

The study of these molecular alterations becomes even more intriguing when they are examined in connection with histological subtypes. Krings and Chen (2018) found variation among the MpBC subtypes (SpCC, SqCC, and MCHMD) [43]. Thus, SqCC and MCHMDs had a more frequent *TP53* mutation, but none of the pure SpCC harbored *TP53* mutations. Furthermore, when different histological subtypes of MpBC were studied, *TP53* mutations were more common in what were described as chondroid matrix-producing carcinomas. *PIK3CA* alterations are higher in MpBCs compared to other TNBCs, and it is more frequently expressed in SpCC. Some authors have demonstrated that patients harboring *PIK3CA* mutations had significantly worse recurrence-free and overall survival outcomes [54]. Interestingly, chondroid-matrix producing carcinomas lacked *PI3K family* and *TERT* promoter mutations, compared to non-matrix-producing tumors. Moreover, *TERT* alteration was seen in SpCC and SqCC but not in MCHMD [31,43].

The ARTEMIS prospective clinical trial (NCT02276443) evaluated TNBC cases in 211 patients, 39 of which were MpBCs [55]. The patients’ biopsies were analyzed before treatment to recognize TNBC subtypes through RNA sequencing and whole exome sequencing. The purpose was to identify chemotherapy-insensitive triple-negative tumors during doxorubicin-cyclophosphamide–based neoadjuvant therapy and to identify a group of patients who could benefit from personalized treatment based on their individual molecular profiling. Among the various conclusions drawn from the study, it was found that a subset of TNBC cases, which lacked histological evidence of metaplastic elements, exhibited a gene expression profile closely resembling that of MpBC. This subset was categorized as “metaplastic-like” based on their gene signature. Both the metaplastic-like TNBCs and MpBCs were enriched in the Vanderbilt mesenchymal (M) and mesenchymal stem-like (MSL) subtypes and showed the upregulation of gene signatures related to hypoxia and the epithelial-to-mesenchymal transition (EMT). When comparing the pathological complete response (pCR) rates among the groups, it was observed that the pCR rate in the metaplastic-like TNBC group was 31%, while the rate in the non-MpBC TNBC group without metaplastic-like gene expression profiles was 39%, and the rate in the MpBC group was 20%. Notably, event-free, metastasis-free, and overall survival were very similar between patients with metaplastic-like TNBCs and those with MpBCs. These findings highlight the significance of using gene expression profiles to identify subgroups within aggressive non-MpBC TNBCs that exhibit a phenotypic resemblance to MpBCs.

Huang-Chun Lien and colleagues (2023) categorized transcriptomic alterations underlying metaplasia into specific metaplastic components in MpBCs [56]. They conducted a gene expression profiling analysis on 59 micro-dissected samples using the NanoString BC360 Panel. Their study compared NST components and paired metaplastic components with spindle, squamous, matrix-producing, and rhabdoid morphologies. The authors revealed that spindle and rhabdoid morphologies were associated with claudin-low signatures, characterized by the upregulation of genes related to stem cells and EMT, but a low expression or downregulation of genes associated with cell–cell adhesion, nucleosome organization, and cell-cycle regulation. Notably, the expression of specific differentially expressed genes varied between the rhabdoid and spindle phenotypes. Additionally, the matrix-producing component exhibited higher expression of genes related to hypoxia mechanisms, while the four squamous components displayed upregulated genes associated with apoptosis, immune response, and cell adhesion.

Overall, we noticed that these studies were primarily focused on high-grade tumors and did not consider low-grade histological types such as FLMC and LGASC, which are outlined in the current WHO histological classification.

## 5. Tumor Immune Microenvironment

Although chemotherapy remains the primary systemic treatment for MpBC, it is noteworthy that MpBCs show a lower rate of response to chemotherapy compared to other forms of TNBC [57,58].

Within the context of this complex landscape and considering the heterogeneity of TNBC and the absence of clearly defined molecular targets, treatment represents a challenge. Consequently, researchers were prompted to explore the tumor immune microenvironment, which has recently unveiled the potential for novel therapeutic strategies.

Several tumor-immune cell interactions participate in the maintenance of an immunosuppressive microenvironment. Programmed death-ligand 1 (PD-L1) is one of the two ligands of programmed cell death protein 1 (PD-1) and is commonly expressed on the surface of immune cells and can be aberrantly expressed on the surface of malignant cells. PD-L1’s binding to PD-1, a CD8 T-cell receptor, causes co-inhibitory signaling that, in turn, causes the de-activation of tumor-infiltrating lymphocytes (TILs) and promotes tumor progression [59].

In patients with TNBC, a high expression of PD-L1 is observed on tumor-infiltrating immune cells, and this can be the key to inhibiting the anticancer immune process. The expression of PD-L1 on the surface of tumor cells is less evident [60], suggesting that PD-1/PD-L1 inhibitors might be used in the treatment of aggressive, advanced, and/or resistant MpBCs. At this point, some authors have highlighted the implementation of automated PD-L1 quantification along with immunohistochemistry in digital image analysis pathology laboratories following a strict validation process [61].

Joneja et al. (2017) evaluated 72 MpBC cases and found PD-L1-positive tumor cells in 46% [51]. In 70 cases in which PD-1 could be evaluated, the researchers found a median of 22.5 TILs in 10 high-power fields. They grouped the TILs’ PD-1 expression as high- and low-expression and related this to the tumor cells’ PD-L1 expression. Interestingly, in 23% of the cases, having PD-L1 expression in tumor cells coincided with a high PD-1 expression in inflammatory cells. Therefore, the researchers concluded that a group of MpBCs could benefit from treatment with immune checkpoint therapy. Data obtained by Dill et al. (2017) demonstrated similar percentages of PD-L1 expression, in 40% of tumor cells, and PD-1 expression, in 80% of immune stromal cells [62]. However, Zhai et al. (2019), who studied 18 cases [10], did not observe PD-L1 immunoreactivity, demonstrating opposite results to those previously mentioned.

Vranic et al. (2020) evaluated 23 pure primary and metastatic spindle cell carcinomas of the breast and found that 33% expressed PD-L1 in tumor cells, some with diffuse patterns [63]. However, PD-L1 expression in immune cells was observed in only two cases; both were triple-negative spindle cell carcinomas.

Chao et al. (2020) evaluated the PD-1/PD-L1 expression in 60 MpBC cases, and they found that more SqCC (55%) exhibited TILs compared with all subtypes [64]. They also found that 50% of cases had both ductal morphology of NST, and metaplastic components (MpBC), and they concluded that tumors with MpBC components had a higher average percentage of TILs compared with the NST histologic type. PD-L1 expression in tumor cells occurred in 50% of the cases, while PD-L1 expression in TILs occurred in 60%. PD-1 expression in tumor cells occurred in 45% of the cases, while positive PD-1 expression in stromal TILs occurred in 55%. Furthermore, their results showed that cases with CD8-positive TILs also had positive PD-L1 expression in tumor cells and positive PD-1 expression in stromal cells. Moreover, they found that samples with a higher number of stromal TILs and CD8-positive TILs were associated with longer disease-free survival. Positive expression of PD-L1 in tumor cells (≥1%) and in stromal cells (≥1%) were also associated with longer survival.

Lien and colleagues (2021) evaluated sTIL abundance and PD-L1 expression in MpBC and found that 34% of cases were intermediate- or high-sTIL-positive [65]. While evaluating this expression in each MpBC subtype, they found that the rate of intermediate or high sTIL was highest in SqCC (50%), followed by MMC (34%), SpCC (30%), and MCHMD (14%). Moreover, when they compared metaplastic components, the matrix-producing/chondroid component of MCHMD and MMC tumors had the lowest rates of intermediate or high sTIL (0% and 7.1%, respectively), along with the osseous component of MMC (0%), compared to the squamous component in SqCC and MMC (50% and 35%, respectively), which showed the highest frequency. A high sTIL infiltration was present only in the carcinomatous components of SqCC and MMC but not in the sarcomatous components (including the spindled, chondroid, osseous, and rhabdoid components). The rate of immune cell PD-L1 positivity was higher than tumor cell PD-L1 positivity for all MpBC subtypes, being highest in SqCC (34%) and lowest in the chondroid component (14% in both MCHMD and MMC). In contrast, tumor cell PD-L1 positivity was consistently lower than immune cell PD-L1 positivity for all metaplastic components.

Yam et al. (2022) found that untreated MpBCs lacked PD-L1 expression in TILs compared with non-MpBC TNBCs [55]. They concluded that the immune checkpoint blockade may only be effective in a small subset of patients with MpBCs.

Appendix A lists the PD-L1 expression in the MpBC case series.

In addition to evaluating the expression of PD-L1 in tumor cells, Kalaw et al. evaluated the FOXP3 (a key marker of T regulatory cells) in TILs of MpBCs [66]. They found that 73% of the samples had PD-L1-positive tumor cells, and 63% had PD-L1-positive TILs. FOXP3 had a positive expression in 73% of sTILs and 57% in intra-tumoral TILs (iTILs). They demonstrated that MpBC presents a differential expression of immune-regulatory markers PD-L1 and FOXP3 with statistical significance compared to other TNBCs. These findings hold promise for the potential use of emerging immunotherapies that target PD-L1 and immunosuppressive T regulatory cells in patients diagnosed with MpBC.

In the context of breast cancer, the PTEN/PI3K and Ras-MAPK pathways play a role in immune evasion and the regulation of PD-L1 expression. As mentioned earlier, *PIK3CA* mutations are notably more prevalent in MpBC compared to other TNBCs, which could account for the significantly heightened PD-L1 expression within tumor cells [45]. Additionally, other factors may contribute to tumoral PD-L1 expression, such as the genomic amplification of 9p24.1 observed in TNBCs [67]. Moreover, the presence of TILs in MpBC, strongly including PD-1-positive TILs as previously described, aligns with the molecular characteristics of these tumors [45].

EMT induction (another remarkable feature of MpBC) can also have an influence in the PD-L1 expression by upregulating its expression. ZEB1 and miR-200 are associated with EMT-activated human breast cancer cells (see Section 6). Furthermore, a meta-analysis by Dong et al. revealed that the intrinsic PD-L1 of tumor cells evidently contributes not only to EMT, but also to the tumor invasion properties, and tolerance to chemotherapy in various tumor types [68].

Using the NanoString, Lieu et al. (2023) demonstrated that cases with prominent spindle components showed an enrichment of macrophage signatures, and the upregulation of genes *PD-L2* (also called *PDCD1LG2*) and *B7-H3* that are associated with immune inhibitory pathways [56]. On the other hand, spindle components showed the downregulation of the *TIGIT* gene, which is related to immune activity. Furthermore, cases with sarcomatous morphology had lower indices of TILs compared to those with carcinomatous morphology. All this led the authors to conclude that carcinomatous and sarcomatous components show different characteristics in the microenvironment. However, they emphasized the need to continue studying this important aspect.

In general, clinical trials have been performed to explore the efficacy, safety, and pharmacokinetics of anti-PD-L1 or anti-PD-1 antibodies in breast cancer. The KEYNOTE-522 clinical trial (NCT03036488) that considered patients with early TNBC found a higher pCR in patients treated with PD-1 antibody followed by adjuvant chemotherapy compared to patients treated with placebo plus chemotherapy [69]. Based on these results, in 2021, the FDA approved pembrolizumab for the treatment of high-risk, early-stage TNBC [70]. Additionally, atezolizumab, durvalumab, and CTLA-4 inhibitors are being studied as possible therapeutic agents. However, these studies are still ongoing [71].

Finally, the finding of MpBC cases showing immune cell PD-L1/PD-1 positivity indicates that cases that typically respond poorly to chemotherapy, and a certain type of MpBC, may benefit from immune checkpoint inhibitor therapy.

## 6. Epithelial–Mesenchymal Transition

EMT is a process characterized by the promotion of cellular changes from an epithelial to a mesenchymal phenotype. Epithelial cells lose their characteristic polarity and gain features related to high aggressiveness, progression, migration/invasion capability, prevention of cell death and aging, and, most importantly, poor response to treatment and resistance to chemotherapy. This phenotypic plasticity is observed in MpBCs previously classified into the basal-like tumors group, which are characterized by a strong claudin-low/EMT-enriched phenotype. Thus, this epithelial–mesenchymal ability in MpBCs is responsible for the tumor progression, intratumoral heterogeneity, metastasis capability, and therapeutic resistance observed in these particular tumors [72,73].

For these reasons, a growing list of EMT regulators has been studied and identified, including extracellular and transcription factors, microRNAs, and the microenvironment [56]. Although cancer stem cell properties and EMT have been recognized by many studies, evidence leads to the possibility of there being EMT-independent mechanisms with different EMT inducers [74].

Much research has been carried out in this regard, and, in this context of a highly dynamic process, many regulatory component pathways have been discovered to play key roles in cancer progression related to EMT. González-Martínez et al. described the pathways TGF-β, canonical WNT, and NOTCH as the most important pathways related to EMT [75]. These authors brilliantly presented the most frequently altered genes in MpBC in Figure 3 of their review paper [75].

Below, we summarize the main and most notable aspects in the general context of this intricate process.

### 6.1. Cadherins and Cadherin-Switching Mechanism

It is known that a switching from E-cadherin to N-cadherin and cadherin 11 is implicated in the process of gaining a mesenchymal profile. This mechanism is associated with the processes of down- and upregulation. Among the epithelial markers downregulated are E-cadherin, occludens zone 1 (ZO1), desmoplakin, and some keratins [40] while EMT-transcriptional inducers SNAI1, SNAI2, and ZEB1 and ZEB2 (E-cadherin repressors) are overexpressed. As expected, genes associated with the mesenchymal phenotype are also upregulated in MpBC. However, E-cadherin loss seems to be necessary but is not the only mechanism implied in developing a complete EMT process [75]. In this context, it is important to remember that other breast tumors are typically characterized by this distinctive alteration, and they do not seem to have been related to EMT mechanisms.

### 6.2. Epithelial and Mesenchymal Markers

The expression of EMT markers through immunohistochemistry is a biological hallmark in MpBC. In this context, EMT markers have been linked to elevated expression of CD44/CD24 and CD29/CD24 ratios, as well as ALDH-1 expression, indicating the presence of stem-cell-like cells within the cellular composition of MpBC.

Britta Weigelt et al., in 2015, evaluated the intrinsic subtypes [41]. They found that claudin-low subtype MpBCs had spindle metaplasia and showed the downregulation of EMT-related genes. They also found that basal-like or normal breast-like subtypes MpBCs had squamous or chondroid metaplasia. Then, in 2017, Piscuoglio et al., in their EMT-related genes analysis [46], concluded that CDH1 and EPCAM were differently expressed in SpCC tumors compared with MpBCs with squamous and chondroid components. Another gene with a role in bone formation, the oncostatin M receptor gene (*OSMR*), was found to be significantly more frequently mutated in the sample of MpBCs evaluated by Beca F et al. (2020) with osseous differentiation, but these authors acknowledged their small sample size [76].

### 6.3. Epithelial–Mesenchymal Transition—Transcriptional Factors

The existence of a network of transcriptional factors that directly represses epithelial genes and upregulates mesenchymal genes is well-known. When this network of EMT-Transcriptional Factors (EMT-TFs) is upregulated, it has an important role in the transcription regulation of EMT. These EMT-TFs are represented by members of the SNAIL, TWIST, and ZEB1 families [77,78,79,80]. These factors interfere in cancer progression at different stages, as well as in their invasion capability, cell-cycle regulation, and drug resistance to multiple therapeutic strategies. However, how these factors intervene in some of this process has not been completely clarified [81].

Tran et al. reported that SNAIL1 expression in primary mouse breast tumors was a strong predictor for the production of disseminated tumor cells (DTCs) and subsequent overt lung metastases [82].

The upregulation, methylation of *TWIST1*, and hormone treatment resistance in breast tumors have been reported [83,84,85,86,87]. Elevated Twist expression was associated with increasing nodal involvement and with patients who had died because of breast tumor. [85].

In 2022, Zhang et al. presented a systematic review and meta-analysis that evaluated the Slug (SNAIL2) protein expression in breast cancer. They concluded that increased Slug protein expression is associated with poor OS (overall survival) and DFS (disease-free survival) [88].

Jang et al. demonstrated that ZEB1 expression was related to MpBC cases, and to poor DFS. They also found that the different histologic variants of MpBC and Zeb1 expression were an independent prognostic factor of DFS [89]. Furthermore, a positive and significant correlation has been established between ZEB1 expression and the clinicopathological characteristics of MpBC. This connection suggests that ZEB1 is expressed in tumors displaying mesenchymal differentiation, including spindle, chondroid, or osseous components, such as FLMC. However, Zawati et al. (2022) reported that LGASC tumors did not exhibit positivity for ZEB1 [20].

### 6.4. Hypoxia

Hypoxia has been described as an important stimulating factor involved in cancer development and the promotion of metastasis through hypoxia-induced factors 1 and 2 (HIF1 and HIF2) [90]. The authors have shown that hypoxia regulates gene expression [91], affects the TME [92], favors dysfunctional vascularization, and promotes the acquisition of the EMT phenotype [93], resulting in the ability to carry out cell mobility and metastasis. These events lead to treatment resistance [94].

Yam et al. (2022) found that hypoxia and EMT gene signatures were upregulated in MpBCs, which indicates a divergence from an epithelial phenotype and enrichment of EMT in these tumors [42,55,95,96]. It has been demonstrated that enrichment in EMT is correlated with poor survival [96].

### 6.5. miRNAs

E-cadherin transcriptional repressors with functions of EMT-transcriptional inducers (SNAI1, SNAI2, ZEB1, and ZEB2) are not the only EMT drivers. The miR-200 family has an important role in the maintenance of an epithelial phenotype [18,97,98]. Epigenetic regulation also seems to be implicated [99,100].

### 6.6. Gene Expression

Lien et al. (2007) evaluated four cases and classified them into two groups according to their high or low carcinomatous components and sarcomatous components. The sarcomatous components also included those associated with ECM formation and the presence of osteochondroid changes, both associated with EMT [42]. Using SAM analysis, they evaluated the epigenetic landscape and compared these two groups. Thirty-five genes were overexpressed in MpBC with predominantly spindle components. They describe their findings explaining that “Twenty-seven of the 35 genes had known biological functions and most of the 27 genes were categorized into six biological processes according to Gene Ontology annotations: cell/cell matrix adhesion, development, signal transduction, metabolism, ion transport, and transcription. Many of the genes were functionally related to ECM remodeling/synthesis (*ADAMT5, HTRA3, MXRA8,* and *TIPM3*), adhesion (*AGC1, EDIL3,* and *PKD2*), and structural constituent/matricellular proteins (*COL16A1*, *COL18A1*, *LUM*, *P4H8*, *SPARC*, *THB1*, and *THB2*). Several other genes had functions related to development (*HOXA7, MSX1, POSTN, AGC1, PRRX1, SFRP2, SFRP4,* and *TBX2*), especially musculoskeletal/bone/cartilage development (*MSX1, POSTN, AGC1, PRRX1, SFRP2,* and *SFRP4*). Genes involved in signal transduction (*PDGFA* and *PDGFRA*), ion transport (*KCNE4* and *PKD2*), cell division (*STAG2*), and muscle contraction (*PPP1R12A* and *TPM4*) were also among the genes identified by SAM analysis”.

Thus, EMT in MpBC is characterized by the loss of intercellular adhesion (E-cadherin and occludins), downregulation of epithelial makers (cytokeratins) and underexpression of genes with biological epithelial functions, upregulation of mesenchymal markers (vimentin and SMA) and overexpression of genes with biological mesenchymal functions, acquisition of a fibroblast-like (spindle) morphology with cytoskeleton reorganization, and increased motility, invasiveness, and metastasis capabilities [101,102,103].

EMT has also been demonstrated in circulating tumors cells (CTC). Bulfoni et al. analyzed metastatic breast carcinoma (blood sample CTCs), and determined that the detection of mesenchymal and/or epithelial mRNAs is related to disease progression. This assessment could serve as an early marker of a response to systemic therapy leading to individualized targeted treatments [104].

Finally, several signaling pathways (those described as primary mediators TGF-b, Notch, and Wnt), hypoxia-related factors, and the expression of miRNAs participate in the regulation of EMT process [105].

## 7. Treatment and Prognosis

According to the NCCN Clinical Practice Guidelines in Oncology, the treatment approach for MpBCs aligns with that of invasive ductal or lobular carcinomas of the breast [106]. The treatment regimen typically involves surgery, adjuvant or neoadjuvant chemotherapy, and radiation therapy, with consideration given to factors such as hormone receptor status and TNM stage at the time of diagnosis [19,106]. Nevertheless, other therapeutic options with well-studied and established targets should be considered, such as, for example, platinum based treatment, immunotherapy, and chemotherapy in MpBC [75] [107,108]. Gadaleta-Caldarola et al. suggested the use of liquid biopsy as a useful tool to predict prognosis, to evaluate the effectiveness of targeted therapy and even the detection of the development of resistance [109].

Many studies have confirmed MpBC is an invasive breast carcinoma with poor prognosis compared with conventional invasive ductal carcinomas and other TNBCs. Series comparing patients with MpBC and non-MpBC TNBC have shown statistically significant differences regarding five-year survival rates, reaching up to 90% for TNBC and 65% for MpBC [22]. The rate of three-year breast cancer–specific survival (BCSS) was 83% for TNBC and 78% for MpBC, and the three-year overall survival (OS) rate was 80% for TNBC and 74% for MpBC [110]. When three-year BCSS was compared between triple-negative MpBC and non–triple-negative MpBC, the rate was 77% and 80%, respectively, and the three-year OS rate was 73% in triple-negative MpBC and 76% in non–triple-negative MpBC. Therefore, triple-negative MpBC has a worse prognosis than non–triple-negative MpBC both in BCSS and OS, while the BCSS and OS of non-triple-negative MpBC were not statistically different from those of non-MpBC TNBC. A similar conclusion was obtained by comparing the five-year survival rates of patients with MpBC (54–88%) and non-MpBC (85–98%), reported as statistically significant differences [22].

Interestingly, significant differences were observed in disease-free survival between patients with MpBC and those with other invasive breast cancer subtypes included as one group, but no significant differences were found in OS between MpBC and each individual invasive breast cancer subtype [22,111]. Corso et al. considered a total of six studies in their meta-analysis, with more than 59,000 patients, and found that MpBC have a worse OS prognosis compared to TNBC with no special type and a significant higher risk of death of 40% [108]. However, they did not find any significant differences in DFS in these two groups. In order to understand these different results, they suggested that DFS could be considered a substitute of OS in TNBC cases based on the fact these tumors have been reported to demonstrate recurrence early, which affects more the DFS.

Rakha et al. (2015) reported the prognosis of high-grade MpBC by comparing different histological subtypes [24]. They concluded that the matrix-producing and SqCC subtypes were associated with a better prognosis compared with the SpCC and mixed spindle and squamous subtypes. McCart Reed et al. (2019) reported that, when considering the three prominent MpBC types (Mixed MpBC, SpCC, and SqCC), the prognosis is worst in Mixed MpBC and SpCC [112]. However, some series with a small number of cases and different histological subtypes did not conclude that there could be statistical significance between the subgroups. In 2009, Downs-Kelly et al. evaluated the prognosis of 32 matrix-producing carcinoma cases. They divided the percentage of matrix elements into three groups (≥10%, >10% and <40%, and ≥40%), demonstrating that tumors with a ≥40% presence of matrix elements showed better DFS survival [113]. Finally, as a conclusion regarding the prognosis among the different histological components in MpBC, we can conclude that every subtype has demonstrated different behavior, reflected in the prognosis and response to treatment, and, in general, these MPBCs subtypes are more aggressive, with higher distant tumor recurrence rates compared to invasive ductal carcinoma [108].

Recently, a study evaluating the OS of patients with different stages of MpBC revealed that the median OS was 69 months for patients with stage I disease, 103 months for stage II, 38.5 months for stage III, and 19 months for stage IV [19].

In some series, low-grade MpBC subtypes occurred at a very low frequency, and, thus, these subtypes were not considered.

Receptor status is another crucial factor for assessing the prognosis of MpBC. One study did not identify significant differences in prognosis between MpBC cases with triple-negative and non-triple-negative receptor statuses [102]. In contrast, another study reported a poorer prognosis for triple-negative MpBC cases compared to MpBC cases with ER positivity. However, no significant differences in prognosis were found between triple-negative MpBC cases compared to those with HER2-positive MpBC [114].

Chemoresistance in TN MpBC and non-TNBC MpBC tumors has been described as higher compared with other TNBCs, which gives MpBCs a worse prognosis. Studies evaluating the pathological response of 150 patients with MpBC who received neoadjuvant chemotherapy with anthracycline and taxane regimens have found an 11% complete pathological response rate [12,22,39,111,115,116,117,118,119,120,121,122]. However, Cortazar et al. (2014), in a meta-analysis of 12 clinical trials, found a higher frequency of complete pathological response that reached up to 33% in patients with TNBC [123].

A prolonged disease control in metastatic disease and pCR have been found in many studies evaluating different types of agent classes and determining the effect of these agents in MpBCs histological variants [124,125,126,127,128,129,130]. Thomas et al. summarize these some “exceptional” responses in patients with MpBC using immunotherapy and immunotherapy combinations, PARP Inhibition with deleterious germline BRCA mutations, anti-angiogenesis, pathway Inhibition, and BRAF inhibition/MEK inhibition combination [131].

## 8. Clinical Trials

At the time of writing this review, we found eight clinical trials in which patients with MpBC were included. We did not find trials exclusively tailored for MpBCs, most likely due to their infrequency and lack of targeted therapy. In fact, one such trial was terminated due to poor enrollment (NCT04549584) [132]. Disease types included in the six trials included all types of breast cancer, advanced and metastatic cases, HER2-negative breast cancer, TNBC, invasive breast carcinoma, and more than 90 different conditions. Interestingly, the results obtained in the KEYNOTE-522 clinical trial (NCT03036488) lead to the FDA approval of Pembrolizumab for the treatment of high-risk, early-stage TNBC [70].

Appendix A lists the eight clinical trials in which cases of MpBC have been included.

## 9. Conclusions

MpBC, an infrequent and notably heterogeneous malignancy, presents a wide range of histological variations and is characterized by a high degree of aggressiveness. It exhibits diminished responsiveness to traditional chemotherapy and an overall prognosis that is less favorable compared to other subtypes of TNBC and invasive breast carcinoma. These unique features can be attributed to the exceptional phenotypic adaptability seen in its molecular peculiarities, the immunological microenvironment, and the epithelial–mesenchymal features it holds.

Due to its aggressiveness and resistance to conventional therapies, the management of MpBC is particularly challenging. Unfortunately, only a few clinical trials have been dedicated to addressing this distinctive cancer subtype.

As ongoing research continues to uncover the complexities of the immune microenvironment and the EMT process in MpBC, there is optimism for the emergence of new therapeutic strategies that could broaden the spectrum of treatment options and, ultimately, enhance the prognosis for MpBC patients. The expanding pool of molecular data and the emergence of promising novel treatment insights emphasize the need for further research initiatives and additional clinical trials with the aim of improving patient survival.

## 10. Limitation of the Study and Current Challenges

There are many limitations when MpBC is studied in its several aspects. The manifest heterogeneity of these tumors explained and demonstrated in all aspects (clinical–pathological, histopathological and immunohistochemical, molecular aspects, immunological microenvironment, EMT, and response to treatment and prognosis) provides these tumors, in particular, complexity in their diagnosis and translational aspects.

First, throughout history, authors have used different definitions of metaplastic carcinoma. This challenge could be overcome if we stick to the current and accepted definition presented in the current WHO classification. Although a common overlap of histological types with each other is expected, we consider that each histological type must be separated (even if they are found in the same patient/sample), to characterize each variant subtype specifically. In this sense, marker expression, clinical behavior, and prognosis reports and outcomes could be analyzed by being separated in each subtype, which allows us to have a better comprehension of these malignances.

Unfortunately, only a few clinical trials have been dedicated to addressing this distinctive cancer subtype. Some facts may explain this reality. One of them is that an adequate number of eligible patients is not obtained. For this, according to the Rare Cancers Europe (RCE) Consensus Panel, “an option is to carry out low-power randomized clinical trials” [133,134]. Another explanation could be the small number of cases feasible. This could be addressed by the establishment of collaborative alliances that involve health centers and organizations between countries and continents.

Moreover, series in which low-grade MpBCs were analyzed are few, and, within them, the number of these cases is small, which may affect the results in terms of statistical significance. The authors of the present article would like to propose a retrospective re-evaluation of breast tumors with an adenosquamous carcinoma low-grade component and with a similar histologic morphology to breast fibromatosis.

## 11. Future Directions and Recommendations

Newly emerging technological advancements, particularly in digital image analysis (DIA) and artificial intelligence (AI), offer promising avenues for enhancing the histologic classification of this entity. These tools enable a thorough analysis and exploration of the data, uncovering intrinsic characteristics and facilitating the discovery of associated predictive and prognostic factors.

Furthermore, cutting-edge high-throughput technologies provide in-depth observations and associated data, which are instrumental in guiding novel treatment approaches. These advancements signal a new era in understanding and treating the condition [75,135,136,137,138,139,140].

Given the limited number of dedicated clinical trials for this disease, it is imperative that we raise awareness within the scientific community and advocate for the translation of new discoveries into actionable treatment options.

## Figures and Tables

**Figure 1 cancers-16-01433-f001:**
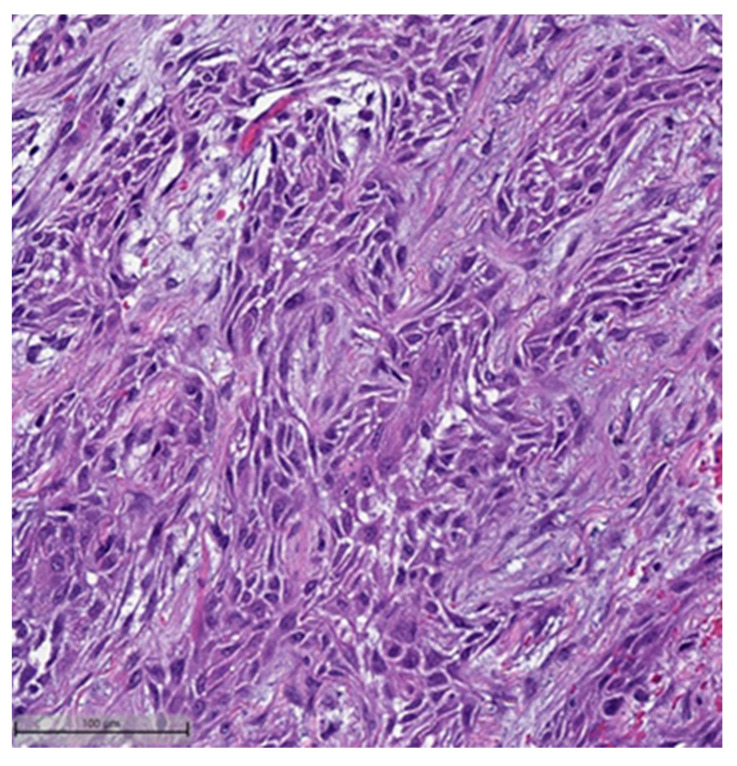
Squamous cell carcinoma component. This case also showed myxoid matrix areas (H&E, 20×).

**Figure 2 cancers-16-01433-f002:**
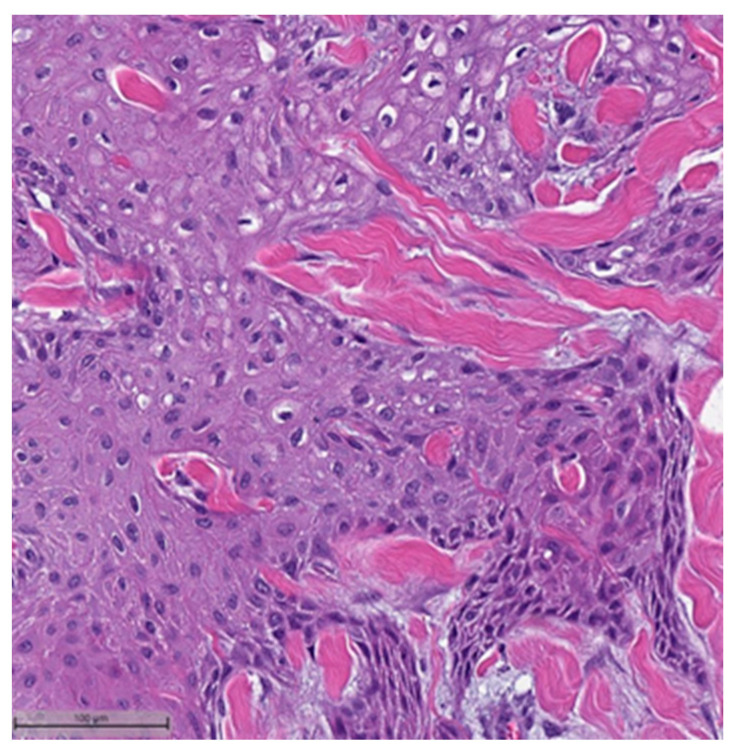
Squamous cell carcinoma component (H&E, 20×).

**Figure 3 cancers-16-01433-f003:**
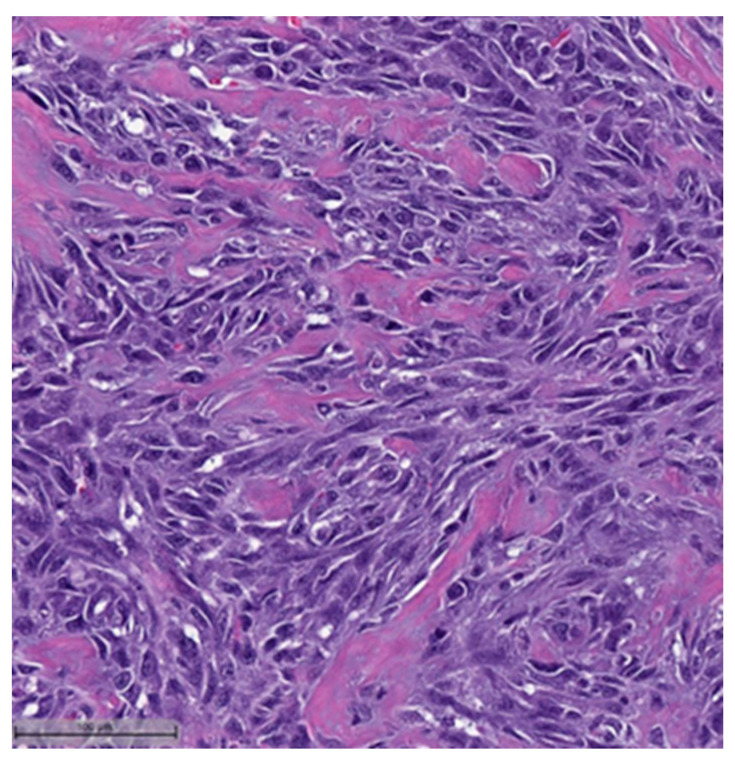
Spindle cell carcinoma (H&E, 20×).

**Figure 4 cancers-16-01433-f004:**
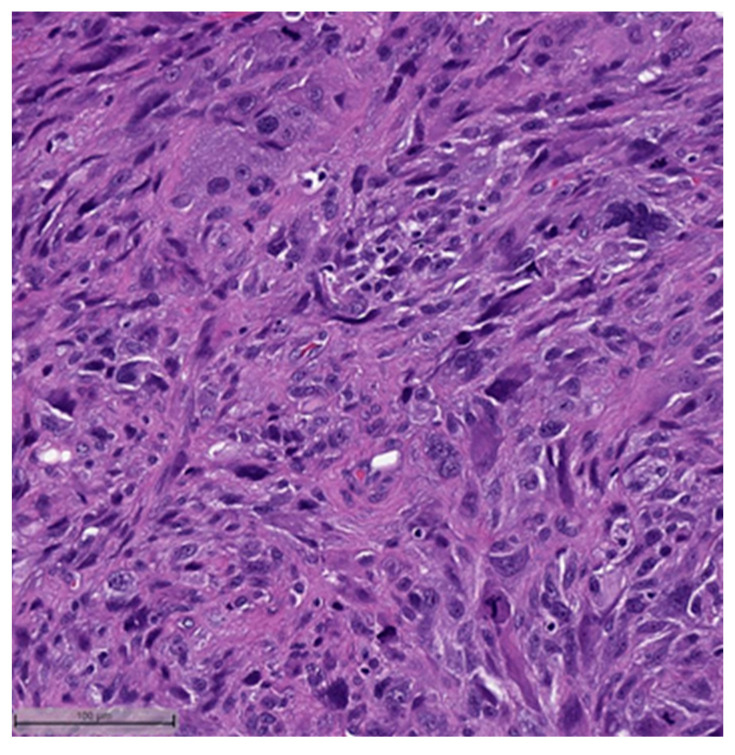
Spindle cell carcinoma component with pleomorphic features (H&E, 20×).

**Figure 5 cancers-16-01433-f005:**
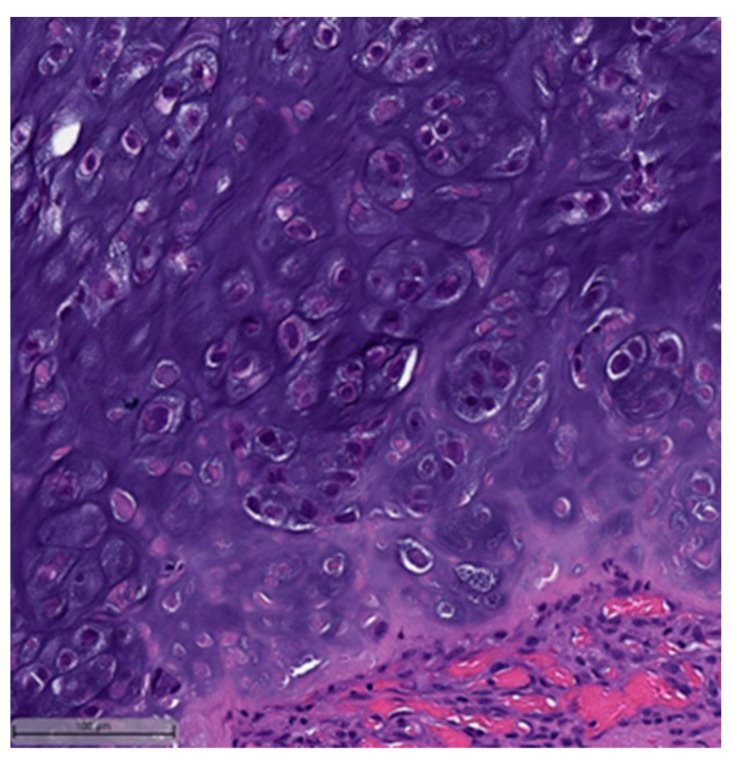
Metaplastic breast carcinoma with chondroid differentiation (H&E, 20×).

**Figure 6 cancers-16-01433-f006:**
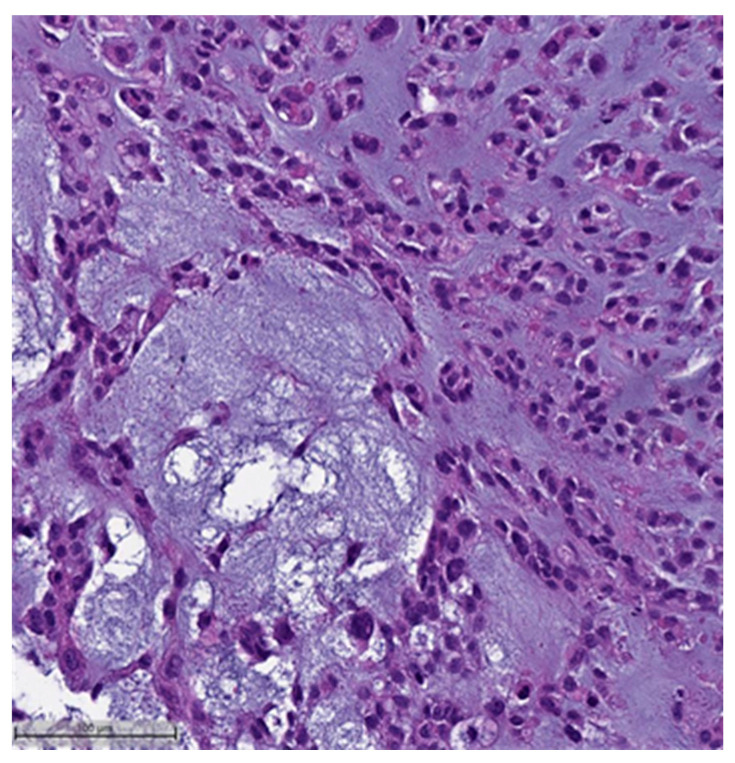
Metaplastic breast carcinoma with matrix-producing component (H&E, 20×).

**Figure 7 cancers-16-01433-f007:**
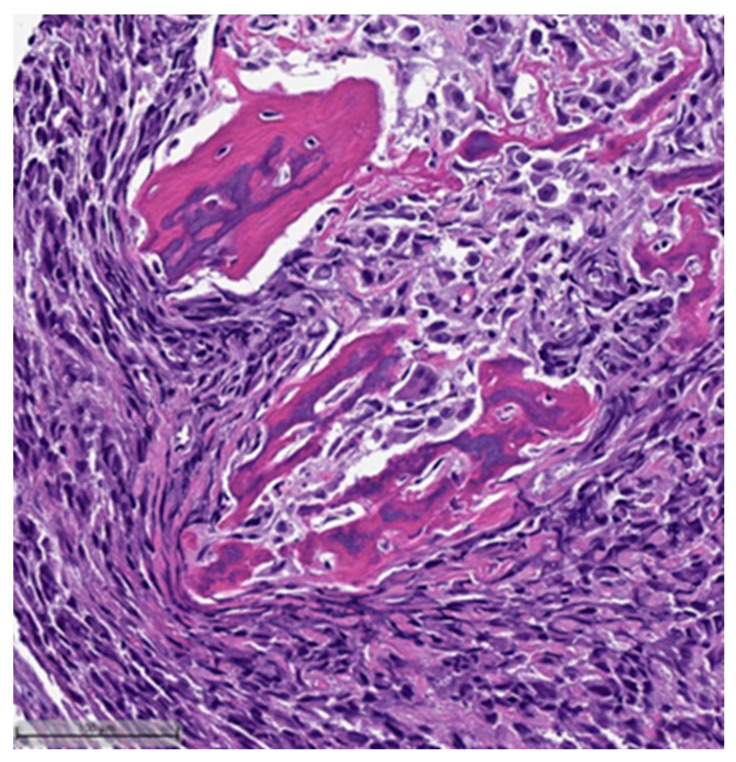
Metaplastic breast carcinoma with osseous differentiation (H&E, 20×).

**Figure 8 cancers-16-01433-f008:**
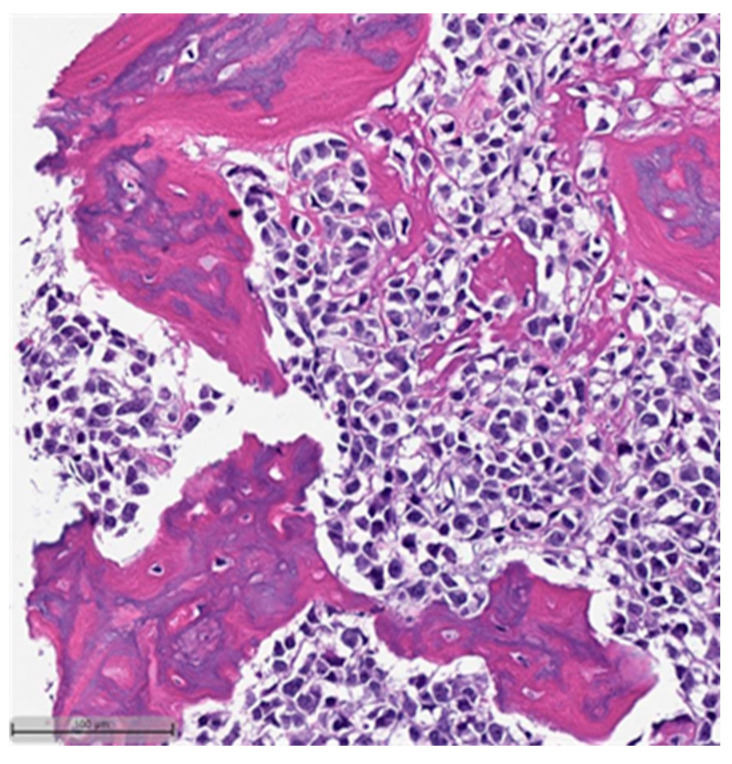
Same case in Figure 7: showing extensive obvious bone trabeculae and hematopoietic tissue (H&E, 20×).

## Data Availability

Not applicable.

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
