# Peer review of "Translational Aspects in Metaplastic Breast Carcinoma"

_cancers, 2024, doi:10.3390/cancers16071433_

Round 1
Reviewer 1 Report
Comments and Suggestions for Authors
Many thanks for submitting this excellent comprehensive review regarding metaplastic breast cancer. To maximise the impact of this review, I would like to suggest the authors to discuss in details: 1)The limitation of the study and the current challenges 2) Future directions and recommendations.
Comments on the Quality of English LanguageNo major issues.
Author Response
Dear Reviewer,
Thank you for your insightful and encouraging feedback on our manuscript titled "Translational Aspects in Metaplastic Breast Carcinoma". We sincerely appreciate your time and effort in reviewing our work. We are grateful for your constructive suggestions to enhance the impact of our review. Your recommendations regarding the discussion of the limitations of our study and the current challenges, as well as proposing future directions and recommendations, are invaluable for strengthening the comprehensiveness of our manuscript.
To address your suggestions:
- Limitations of the Study and Current Challenges: We acknowledge that despite our efforts to provide a comprehensive overview of metaplastic breast cancer, there are inherent limitations to our study. We emphasized these limitations in the revised manuscript, including potential biases in the selection of studies, gaps in the available literature, and any methodological constraints that may have influenced our findings, in lines 587-612. Additionally, we elaborated on the current challenges faced in the diagnosis, treatment, and management of metaplastic breast cancer, considering factors such as disease heterogeneity, limited therapeutic options, and prognostic uncertainties.
- Future Directions and Recommendations: In response to your suggestion, we expanded the discussion section to include insights into future research directions and practical recommendations. This encompasses areas such as the exploration of novel biomarkers for early detection, advancements in personalized treatment strategies tailored to the molecular profile of metaplastic breast cancer subtypes, and the evaluation of emerging therapeutic modalities, including immunotherapy and targeted therapies. Furthermore, we provided insights regarding the importance of multidisciplinary collaboration, patient education, and ongoing clinical trials in optimizing patient outcomes in lines 613-625.
Once again, we express our gratitude for your valuable feedback, which will undoubtedly contribute to the refinement of our review.
Sincerely,
Maria G Raso MD
Elizve Barrientos-Toro MD
Reviewer 2 Report
Comments and Suggestions for Authors
In this manuscript, the authors have provided a collection of information on the translational aspects of metaplastic breast carcinoma. This is an interesting topic. The following points should be addressed before considering for publication.
1. A separate section on the molecular signaling networks of metaplastic breast carcinoma and their translational relevance should be included.
2. A table or figure on the molecular alterations in metaplastic breast carcinoma will improve the manuscript.
Comments on the Quality of English LanguageModerate English language editing is needed.
Author Response
Dear Reviewer,
We greatly appreciate your thoughtful review of our manuscript titled "Translational Aspects in Metaplastic Breast Carcinoma". Your insights are invaluable in refining our work to ensure its relevance and comprehensiveness. We have carefully considered your comments and suggestions, and we are committed to addressing them effectively.
Inclusion of a Section on Molecular Signaling Networks: We acknowledge the importance of incorporating a dedicated section on the molecular signaling networks of metaplastic breast carcinoma and their translational relevance. In the revised manuscript, we introduced a comprehensive discussion elucidating the key molecular pathways implicated in the pathogenesis, progression, and therapeutic response of metaplastic breast carcinoma. This section highlights the pivotal roles of signaling cascades such as the Her2 pathway (lines 128 and 168-169), TP53 pathway (lines 170-172 and 200-204), the WNT pathway (lines 175-176, 374-376 and 481-483), the PI3K/AKT/mTOR pathway (line 186-188, 207-208 and 323-326), the Ras-MAPK pathway (lines 183 and 323-324), the PD-L1 pathway (lines 255-260), the PD-L2 pathway (lines 337-340), the B7-H3 pathway (lines 337-340), pathways related to EMT (lines 374-377), network of EMT-Transcriptional Factors (EMT-TFs) (lines 408-410), among others, in driving the phenotypic heterogeneity and therapeutic resistance observed in metaplastic breast carcinoma. Additionally, we emphasized the clinical implications of targeting these molecular alterations in the development of novel therapeutic strategies and predictive biomarkers for personalized treatment approaches.
Inclusion of a Table or Figure on Molecular Alterations: We appreciate your suggestion to incorporate a visual representation of molecular alterations in metaplastic breast carcinoma to enhance the clarity and accessibility of the manuscript. In response to this feedback, we included a comprehensive tables (Supplementary Tables 1, 2 and 3) summarizing genetic mutations identified in Metaplastic Breast Carcinoma and its frequencies based on compiled publications revised for this review. Regarding the molecular alterations, we referenced in lines 374-377 a publication that brilliantly summarized and presented this topic as a Figure: González-Martínez et al “Epithelial Mesenchymal Transition and Immune Response in Metaplastic Breast Carcinoma” (Ref. 75).
Thanks for suggesting these enhancements to our manuscript to enrich its content and facilitate the dissemination of knowledge on the translational aspects of metaplastic breast carcinoma. Your feedback has been instrumental in guiding us towards improving the quality and impact of our work, and we sincerely thank you for your valuable input.
Sincerely,
Maria G Raso MD
Elizve Barrientos-Toro MD
Reviewer 3 Report
Comments and Suggestions for Authors
This is a review of the clinicopathological and molecular features of metaplastic breast carcinoma. The main limitation of this review is that the authors do not highlight which are the main new findings they reported in comparison with many other recent reviews since 2020 (PMID: 36696934, PMID: 37169444, PMID: 37179225, PMID: 37528884, PMID: 37218987 , PMID: 35236632, PMID: 35088973, PMID: 34299016, PMID: 33538176 , PMID: 32650408). Whereas some of them are cited, other are not; for example PMID: 35088973 is a systematic review and meta-analysis focus on prognosis, and PMID: 34299016 is focus on EMT and the immune microenvironment, two sections of present Review.
In general, the study is poorly structured. It is more reasonable to discuss first EMT, which is the main characteristic of MBC, and then other features such as immune response. The description of the molecular features should also be ordered according the type of study/findings: mutations, trnascriptomics, etc.
Author Response
Dear Reviewer,
Thank you for your detailed review of our manuscript titled "Translational Aspects in Metaplastic Breast Carcinoma." Your feedback is invaluable in helping us improve the clarity, structure, and relevance of our review. We appreciate the opportunity to address your concerns and enhance the quality of our work.
Highlighting Main New Findings in Comparison with Recent Reviews: We acknowledge the importance of highlighting the main new findings reported in our review compared to other recent publications on metaplastic breast carcinoma. In the revised manuscript, we provide a clear delineation of the novel contributions and insights offered by our review, particularly in comparison to the articles cited, such as PMID: 36696934 (lines 69-70, Ref. 21), PMID: 37169444 (lines 124-125, Ref. 28), PMID: 37179225 (lines 554-557, Ref. 132), PMID: 37528884 (line 64, Ref. 20), PMID: 37218987 (lines 64, 487-489 and 534-536, Ref. 19), PMID: 35236632 (lines 124-125, Ref. 27), PMID: 35088973 (lines 512-515 and 529-533, Ref. 108), PMID: 34299016 (lines 374-376, 387-388, 489-492 and 619-622, Ref. 75), PMID: 33538176 (lines 492-494, Ref. 120), PMID: 32650408 (lines 71-74, 170-172, 186-188, 191, 496-499, 506-512, 546-549, Ref. 22), and others. We emphasized unique aspects of our review, including the integration of clinicopathological and molecular features, the comprehensive analysis of epithelial-mesenchymal transition (EMT) and the immune microenvironment, and the synthesis of recent advancements in understanding the molecular landscape of metaplastic breast carcinoma. Additionally, we will ensure consistency in citation and appropriately acknowledge the significance of systematic reviews and meta-analyses, such as PMID: 35088973 (lines 512-515 and 529-533, Ref. 108), in the context of our discussion.
Improving the Structure of the Review: We appreciate your feedback on the structure of our review and agree with your suggestion to reorganize the content for better coherence and flow. In the revised manuscript, we expanded and prioritized the discussion of EMT (lines 364-366, -374-380, 387-391, 406-440, 445, 451, 481-483), considering its central role as a defining characteristic of metaplastic breast carcinoma, followed by an exploration of other relevant features such as the immune response and molecular alterations. Furthermore, we summarized its molecular features (line 151-245) according to the type of study/findings, categorizing mutations, transcriptomics, and other molecular features to provide a more organized and logical presentation of the data.
These revisions ensure that our manuscript meets the highest standards of clarity, rigor, and relevance. Your constructive feedback has been instrumental in guiding us towards enhancing the quality and impact of our review, and we sincerely appreciate your time and expertise.
Sincerely,
Maria G Raso MD
Elizve Barrientos-Toro MD
Reviewer 4 Report
Comments and Suggestions for Authors
The review offers a comprehensive overview of molecular aspects, immune microenvironment, treatment options, and ongoing clinical trials related to MBC. With some minor revisions to enhance clarity and organization, the manuscript will serve as a valuable resource for researchers and clinicians interested in understanding and managing this challenging subtype of breast cancer.
To enhance the clarity and readability of the manuscript, I recommend the following:
Incorporate more recent references to ensure the review reflects the latest research findings in the field of MBC.
EPIDEMIOLOGY
Siegel RL, Giaquinto AN, Jemal A. Cancer statistics, 2024. CA Cancer J Clin. 2024 Jan-Feb;74(1):12-49. doi: 10.3322/caac.21820. Epub 2024 Jan 17. Erratum in: CA Cancer J Clin. 2024 Feb 16;: PMID: 38230766.
THERAPEUTIC OPTIONS FOR SPECIAL HYSTOLOGY SUBTYPES
Trapani D, et al. Benefit of adjuvant chemotherapy in patients with special histology subtypes of triple-negative breast cancer: a systematic review. Breast Cancer Res Treat. 2021 Jun;187(2):323-337. doi: 10.1007/s10549-021-06259-8. Epub 2021 May 27. PMID: 34043122.
THE ROLE OF PLATINUM SALTS
Gerratana L, et al. Do platinum salts fit all triple negative breast cancers? Cancer Treat Rev. 2016 Jul;48:34-41. doi: 10.1016/j.ctrv.2016.06.004. Epub 2016 Jun 11. PMID: 27343437.
EPITHELIAL-TO-MESENCHYMAL TRANSITION
Bulfoni M, et al. In patients with metastatic breast cancer the identification of circulating tumor cells in epithelial-to-mesenchymal transition is associated with a poor prognosis. Breast Cancer Res. 2016 Mar 9;18(1):30. doi: 10.1186/s13058-016-0687-3. PMID: 26961140; PMCID: PMC4784394.
Consider adding subheadings or bullet points to improve the organization and readability of certain sections, particularly those discussing gene mutations and immune microenvironment.
Define technical terms and abbreviations where necessary to facilitate understanding for readers less familiar with the topic.
Author Response
Dear Reviewer,
Thank you for your thorough and insightful feedback on our manuscript regarding metaplastic breast cancer (MpBC). We appreciate your acknowledgment of the comprehensive overview provided and your valuable suggestions for further enhancing the clarity and readability of the manuscript.
In response to your recommendation to incorporate more recent references, we have updated the manuscript with the following relevant studies:
Lines 33-35, Ref 1
Siegel RL, Giaquinto AN, Jemal A. "Cancer statistics, 2024." CA Cancer J Clin. 2024 Jan-Feb;74(1):12-49. doi: 10.3322/caac.21820. Epub 2024 Jan 17.
Line 489-492, Ref 107
Trapani D, et al. "Benefit of adjuvant chemotherapy in patients with special histology subtypes of triple-negative breast cancer: a systematic review." Breast Cancer Res Treat. 2021 Jun;187(2):323-337. doi: 10.1007/s10549-021-06259-8. Epub 2021 May 27.
Line 489-492, Ref 109
Gerratana L, et al. "Do platinum salts fit all triple-negative breast cancers?" Cancer Treat Rev. 2016 Jul;48:34-41. doi: 10.1016/j.ctrv.2016.06.004. Epub 2016 Jun 11.
Line 476-480, Ref 104
Bulfoni M, et al. "In patients with metastatic breast cancer, the identification of circulating tumor cells in epithelial-to-mesenchymal transition is associated with a poor prognosis." Breast Cancer Res. 2016 Mar 9;18(1):30. doi: 10.1186/s13058-016-0687-3.
Additionally, we have taken your advice to heart regarding improving the organization and readability of certain sections. Subheadings have been incorporated in EMT section to enhance clarity and facilitate easier navigation for our readers.
Furthermore, we have ensured that technical terms and abbreviations are defined where necessary throughout the manuscript to aid readers who may be less familiar with the topic. Importantly, MBC was changed to MpBC over the whole manuscript.
We believe that these revisions have significantly strengthened the manuscript and are grateful for your valuable guidance in this process.
Sincerely,
Maria G Raso MD
Elizve Barrientos-Toro MD
Round 2
Reviewer 2 Report
Comments and Suggestions for Authors
The authors have addressed all the comments and the manuscript was improved by the revision. The manuscript can be accepted for publication in its current form.
Comments on the Quality of English LanguageModerate English language editing will improve the readability.
Reviewer 3 Report
Comments and Suggestions for Authors
No additional comments.